# Supercritical Carbon Dioxide Extraction of Four Medicinal Mediterranean Plants: Investigation of Chemical Composition and Antioxidant Activity

**DOI:** 10.3390/molecules26185697

**Published:** 2021-09-20

**Authors:** Lara Čižmek, Mojca Bavcon Kralj, Rozelindra Čož-Rakovac, Dmitrii Mazur, Nikolay Ul’yanovskii, Marko Likon, Polonca Trebše

**Affiliations:** 1Laboratory for Aquaculture Biotechnology, Division of Materials Chemistry, Ruđer Bošković Institute, Bijenička 54, 10000 Zagreb, Croatia; lcizmek@irb.hr (L.Č.); rrakovac@irb.hr (R.Č.-R.); 2Center of Excellence for Marine Bioprospecting (BioProCro), Ruđer Bošković Institute, Bijenička 54, 10000 Zagreb, Croatia; 3Faculty of Health Sciences, University of Ljubljana, Zdravstvena pot 5, 1000 Ljubljana, Slovenia; mojca.kralj@zf.uni-lj.si; 4Chemistry Department, Lomonosov Moscow State University, Leninskie Gory 1/3, 119991 Moscow, Russia; neodmitrii@gmail.com; 5Core Facility Center “Arktika”, Lomonosov Northern (Arctic) Federal University, nab. Severnoy Dviny 17, 163002 Arkhangelsk, Russia; n.ulyanovsky@narfu.ru; 6Škrlj, d.o.o., Batuje 90, 5262 Črniče, Slovenia; marko.likon@telemach.net

**Keywords:** green solvents, chamomile, yarrow, St. John’s wort, curry plant, square-wave voltammetry, chromatography

## Abstract

With everyday advances in the field of pharmaceuticals, medicinal plants have high priority regarding the introduction of novel synthetic compounds by the usage of environmentally friendly extraction technologies. Herein, a supercritical CO_2_ extraction method was implemented in the analysis of four plants (chamomile, St. John’s wort, yarrow, and curry plant) after which the non-targeted analysis of the chemical composition, phenolic content, and antioxidant activity was evaluated. The extraction yield was the highest for the chamomile (5%), while moderate yields were obtained for the other three plants. The chemical composition analyzed by gas chromatography-high-resolution mass spectrometry (GC-HRMS) and liquid chromatography-high-resolution mass spectrometry (LC-HRMS) demonstrated extraction of diverse compounds including terpenes and terpenoids, fatty acids, flavonoids and coumarins, functionalized phytosterols, and polyphenols. Voltammetry of microfilm immobilized on a glassy carbon electrode using square-wave voltammetry (SWV) was applied in the analysis of extracts. It was found that antioxidant activity obtained by SWV correlates well to 1,1-diphenyl-2-picrylhidrazine (DPPH) radical assay (R^2^ = 0.818) and ferric reducing antioxidant power (FRAP) assay (R^2^ = 0.640), but not to the total phenolic content (R^2^ = 0.092). Effective results were obtained in terms of activity showing the potential usage of supercritical CO2 extraction to acquire bioactive compounds of interest.

## 1. Introduction

Polyphenols are a chemically diverse group of plant secondary metabolites that have various protective and defensive roles. It is estimated that about 2% of all carbon photosynthesized by plants, amounting to about 1 × 10^9^ tons annually, is converted into polyphenolic-related compounds [1]. These versatile molecules are characterized by many positive effects, associated with the prevention of inflammatory, cardiovascular and other diseases. Since they are well known for their antioxidant, antiviral and antimicrobial activity, it is no wonder that large amounts of herbal remedies and pharmaceutical supplements rich in polyphenolic compounds have been developed and are consumed daily [2,3]. Considering their putative beneficial biological effects, the focus in further investigations should be towards the development of new candidates for future pharmaceuticals and not so much on the replacement of existing synthetic drugs [4]. For such reason, the optimization of extraction procedures regarding higher yield with respect to environmental safety (green technologies), and confirmation of their activities are still challenges that need to be overcome.

To obtain the highest possible amount of bioactive compounds from plant biomass, effective extraction procedures are required. However, finding the optimal one, suitable for routine extraction procedures, is from a technological point of view extremely challenging. Hydrodistillation, a conventional extraction technique, fails in the extraction of sensitive compounds due to their thermal degradation [5]. The use of other organic solvents (mostly halogenated) in extraction procedures complying with the green chemistry and green engineering principles [6] is generally avoided. Instead, the application of compressed fluids, among them, supercritical fluid extraction (SFE), showed remarkable results over conventional solvent extraction processes. Supercritical fluids (SCF) reach their critical state when two phenomena occur simultaneously: firstly when it is heated above its critical temperature (T_c_), and secondly when it is pressurized above its critical pressure (P_c_). Carbon dioxide (CO_2_) is widely used as SCF because it is not harmful, inert, chemically stable, inflammable, cheap, abundant, environmentally acceptable, easily separated from extract (solvent-free after depressurization) and possesses moderate critical properties (T_c_ = 31.1 °C, P_c_ = 7.38 MPa) [7,8]. Chemically, it is a non-polar solvent (it lays between pentane and toluene [9], good miscibility with methanol or ethanol used to enhance the solubilization of polar substances. In general, the application of SFE offers higher extraction yields due to the higher penetration power of supercritical fluid into porous solid materials, higher selectivity, efficiency, stability, reproducibility, and suitability due to adjustable solvation power (temperature, pressure) [4,10]. In the high-tech and basic chemical industries, supercritical fluids as a production medium are becoming increasingly studied and appreciated. With the development of technology and materials, supercritical liquids have become promising solvents and reaction media in the pharmaceutical, food, and chemistry industry. For this reason, researchers and industries widely use supercritical extraction for producing bioactive components from vegetable matrices. Understanding the processes behind extraction is crucial for performing efficient extraction because there is big differentiation in nature, origin, and preparation of raw materials [11].

In the extraction of herbals for pharmacological purposes it is particularly important to define targeted components. As can be seen from Figure 1, different extraction conditions lead to different extraction yields of individual chemical components that express certain biological activities. Producing herbs in integrated agricultural areas can result in extracts polluted with pesticides, which tend to concentrate on final products during extraction processes [12,13]. However, some Mediterranean and Alpine areas still represent unpolluted oases. 

Four model plants from families *Hypericaceae* (St. John’s wort) and *Asteraceae* (chamomile, yarrow, and curry plant) are grown in the Mediterranean area and are used in versatile applications in pharmacy and medicine [14]. Chamomile (*Matricaria chamomilla*) is a medicinal herb considered as a source of important bioactive compounds with desirable properties (antioxidant, anti-nociceptive, and anti-cancer activities) [15]. Yarrow (*Achillea millefolium*) is one of the oldest plants used in traditional medicine, while antimicrobial, antihypertensive, antihyperlipidemic, antispasmodic, antidiabetic, antispermatogenic-antifertility, and immunosuppressive activities have been reported [16]. Curry plant *(Helichrysum italicum*) is known for its volatile oil, which displays many biological properties, such as anti-inflammatory [17,18], antiallergenic, antimicrobial [19,20,21], antioxidant [18,22], and antiviral [17,23] properties. The only member of the *Hypericaceae* family, St. John’s wort (*Hypericum perforatum*) contains significant levels of biologically active agents, specifically phenolic compounds with pronounced antioxidant, antifungal, DNA protective, and anticholinesterase activities [24,25,26]. Most recently it has been shown that flowers of the plant genus *Hypericum* are also good photoredox catalysts [27] proving that natural plants with versatile photophysical properties can be used in different areas of research. 

This study aimed to test the efficiency of CO_2_ SFE to get as many bioactive components as possible from four model medical herbs used in the traditional Mediterranean environment. The extraction technique was evaluated in terms of extraction yields. The insight in the phenolic profile of individual herbal extracts was defined by a comparison of two analytical techniques (gas chromatography-high-resolution mass spectrometry (GC-HRMS) and liquid chromatography high-resolution mass spectrometry (LC-HRMS)) enabling the determination of different biological components. The antioxidant potential was defined with three commonly used methods (the total phenolic content—TCP; the radical scavenging assay—DPPH and the ferric reducing antioxidant power assay—FRAP) and one alternative voltammetric technique (square-wave voltammetry (SWV). To correlate total phenolic content (TPC) with antioxidant activity (SWV, DPPH, and FRAP), Pearson’s correlation coefficient was used.

## 2. Results

### 2.1. Plant Extract Yields

Total yields of herbal extracts ranged from 3.75 to 7.20% and are presented in Table 1. Since the extracts from collection vessel 1, containing waxes and heavily soluble lipophilic compounds, were not a part of this study, the detailed analysis has not been performed. The extracts from vessels 2 and 3 gave the highest yield in the case of chamomile (5%). For other samples, extraction yield ranged from 2.5% in the case of the curry plant up to 3.18% in the case of St. John’s wort. For a better understanding of the extraction process, a detailed diagram of the supercritical extraction system is shown in Appendix A.

As shown in Figure 2, the density and solubility are a function of temperature and pressure. Pressure has more influence on solubility than the temperature at the extraction of substances with low partial pressure what was the case in our experiments. However, from experiments, it was seen that yield was rising with the increase of the pressure, which is in connection with the extraction of a wider portfolio of different compounds, for example, waxes and fats. 

### 2.2. Identification of the Organic Compounds

#### 2.2.1. GC×GC-HRMS Results

The GC×GC-HRMS analysis mainly revealed the diterpene and sesquiterpene content in all plant extracts. Identification of semivolatile organic compounds was performed using NIST14 EI spectral libraries and available literature data on the organic content of chamomile, yarrow, St. John’s wort and curry plant [28,29]. Among all four analyzed plant extracts the most interesting and reliable results were obtained for chamomile. Six sesquiterpenes (*cis*-β-farnesene, spathulenol, (−)-bisabolol oxide B, bisabolone oxide A, (−)-α-bisabolol oxide A, matricarin) and 2 spiroethers (*E/Z* 2-(hexa-2,4-diyn-1-ylidene)-1,6-dioxaspiro[4.4]non-3-ene) were the major components of chamomile extract identified by GC-HRMS (Table 2). This nicely correlates to the literature data as these compounds were also claimed as main terpene representatives of chamomile. For other plant extracts identification wasn’t that reliable since the proposed molecular ions, fragmentation pattern and library search results didn’t fit any compounds listed as known components of the yarrow, St. John’s wort, and curry plant. For such compounds, Table 2 includes only proposed molecular formulae.

#### 2.2.2. LC-HRMS Results

Similar to the GC-MS approach the molecular formulae and tandem mass spectra obtained with LC-MS were collated with the possible compounds listed as known and proved components of chamomile, yarrow, St. John’s wort and curry plant [28,29]. The list of detected compounds (Appendix A) was classified into several groups of naturally occurring organic substances: fatty acids and their derivatives, flavonoids and coumarins, functionalized phytosterols, polyphenols, sesquiterpenes and terpenoids, and prenylated phloroglucinols. If the molecular formula obtained by accurate mass measurement with ESI-HRMS coincided with that mentioned in the literature and the tandem mass spectrum contained fragment ions in accordance with the proposed structure, the assignment was done the same as in literature. However, many substances were impossible to clearly identify since there were several isomers present with the same molecular formulae and similar fragmentation patterns in one sample. Such compounds were just assigned with a class. The full list of detected compounds by means of LC-MS is summarized in Appendix A.

Fatty acids and their derivativesThis group of compounds was mainly represented by palmitic, oleic, linoleic, and linolenic acids as well as their hydroxy- and epoxy-derivatives.Flavonoids and coumarinsUsually, the identification and quantitation of flavonoids in natural products is a routine procedure using several instrumental methods (LC-MS, LC-DAD, etc.) [30,31,32]. Since the aim of this work was non-target analysis with tentative identification of all possible components without the use of any standards, the detected flavonoids were assigned up to an isomer without specifying the exact position of the functional groups or radicals. This way identified compounds included trihydroxy-trimethoxyflavone, dihydroxy-tetramethoxyflavone, dihydroxy-trimethoxyflavone and methoxycoumarin.Functionalized phytosterolsA significant group of detected compounds with the number of O-atoms from 3 to 6 and degree of unsaturation 7–9 was assumed to represent derivatives of phytosterols—phytosteroids, which occur in plants and vary in the carbon side chain and a number of double bonds (Figure 3). It wasn’t evident from the spectra which exactly functional and structural groups are present, so the main argument for such class assignment was a large list of related structures during database search (ChemSpider, PubChem).PolyphenolsPolyphenols are a very large group of naturally occurring organic compounds with a variable structure bearing several hydroxyl groups in the aromatic rings. Since it’s not a well-defined class of compounds we have assigned a polyphenol class to molecular formulae with the number of O-atoms from 3 to 9 and degree of unsaturation 4–11. Since mass spectra didn’t allow clearly elucidating the structure, in most cases these compounds were not identified. The only two polyphenols reliably identified were helipyrone and italipyrone in the curry plant extract, as their presence in the plant was found in the literature [29]. Also, three isomers of syringaresinol were found in St. John’s wort and curry plant extracts.Sesquiterpenes and terpenoidsThe group of sesquiterpenes and terpenoids detected with LC-MS mainly contained the most polar compounds, because the least polar were covered by the GC-MS method. Worth mentioning was the fact that this class was identified only by means of (+) ESI, which assumes the absence of relatively acidic protons in the structure, namely carboxylic or phenolic groups. Molecular formulae varied within the number of O-atoms from 1 to 5 and degrees of unsaturation 3–8. Reliable identification was possible similarly to other classes mentioned above if the literature reference was available. Thus, only two sesquiterpenes identified from chamomile extract were identified this way—matricin and matricarin.

Prenylated phloroglucinolsA large group of organic compounds with the number of O-atoms varying from 4 to 6 and degree of unsaturation 8–11 represents polycyclic polyprenylated acylphloroglucinol family also known as PPAP family. The PPAPs commonly consist of a highly oxygenated and densely substituted bicyclo[3.3.1]nonane-2,4,9-trione or bicyclo[3.2.1]octane-2,4,8-trione core with various side chains (prenyl, geranyl, etc.) [33]. Within the PPAP family according to literature data we have identified colupulone, furohyperforin, 33-hydroperoxyfurohyperforin, colupone, adlupone, 8-hydroxyhyperforin 8,1-hemiacetal, furoadhyperforin, hyperforin, adhyperforin, and their various isomers and derivatives. Though the absence of standards or any additional data some of the isomers were possible to recognize by comparing their fragmentation patterns. Figure 4 combines ESI(-) CID mass spectra of compounds with the same [M − H]^−^ ion (*m/z* 551.3728) corresponding to formula C_35_H_51_O_5_. Only three possible compounds with formula C_35_H_52_O_5_ which were previously identified in St. John’s wort were found: furohyperforin, 8-hydroxyhyperforin 8,1-hemiacetal, and oxepahyperforin [29]. As is clear from Figure 4 that only mass spectrum **c** contains a peak at *m/z* 523.3812 corresponding to loss of CO molecule, while the most abundant peak at *m/z* 455.3177 corresponds to the consequent loss of C_5_H_8_ fragment. Comparing the assumed structural formulae (Figure 4), we may conclude that only oxepahyperforin—C_36_H_54_O_5_ would be able to lose CO that easily due to oxabicyclo moiety. The consequent loss of two prenyl (C_5_H_9_) radicals resulting in *m/z* 413.2350 together with formation of *m/z* 383.2230 (C_24_H_31_O_4_) ion may occur only for furohyperforin, due to specific position of the substituents, hence leaving spectrum **b** with most abundant *m/z* 411.2528 arising due to loss of the C_5_H_10_ for 8-hydroxyhyperforin 8,1-hemiacetal.

### 2.3. Determination of Total Phenolic Content (TPC)

In this study, extracts from four medicinal plants obtained by supercritical CO_2_ extraction were analyzed for their total phenolic content (TPC). The distribution of phenolic compounds in chamomile, St. John’s wort, curry plant and yarrow plant extracts by spectrophotometric measurement is depicted in Table 3. 

It can be seen that among the four sample extracts, curry plant contained the highest amount, 5.60 ± 0.03 mg GAE/g extract, followed by St. John’wort (2.7 ± 0.02 mg GAE/g) > chamomile (2.3 ± 0.02 mg GAE/g) > yarrow (0.8 ± 0.02 mg GAE/g). TP content in the curry plant was 2-fold higher (*p* < 0.001) than in St. John’s wort and chamomile, while this content was 7-fold higher when compared to yarrow extract.

### 2.4. Voltammetric Analysis

Before the analysis of sample extracts, gallic acid was used as a polyphenolic standard and analyzed utilizing square-wave voltammetry (SWV) on a glassy carbon electrode (GCE) in the form of dry residue. Figure 5a shows a square-wave voltammogram of 2 × 10^−5^ mol/L solution of gallic acid dry residue on GCE. In acidic conditions (pH 2.5) one sharp peak at 0.388 ± 0.002 V is dominant. By changing the frequency (10–200 Hz), the dependence of peak potential with log *f* is linear with the slope 0.349 V/d.u. The sensitivity of the method was evaluated by studying the influence of gallic acid concentration on the net peak current. The dependence of the net peak current with concentrations of gallic acid dry residue on GCE is shown in Figure 5b. The current response was obtained in the range 1.00–230.00 μmol/L (0.17–391.28 mg/L), but the linearity of the response was obtained in the rather narrow range, from 1.00 μmol/L to 20.00 μmol/L with the correlation coefficient, *r* = 0.991. At higher concentrations, surface saturation occurred causing the drop in peak current.

The accuracy of the method is expressed as a recovery for peak P1 and the obtained value was 101.20 ± 9.38%. The relative standard deviation for 2 × 10^−5^ mol/L gallic acid in the form of a dry residue expressed for a current of peak P1 was 7.42% and for potential was 0.59%. The limits of detection (LOD) and quantification (LOQ) for peak P1 were calculated from the parameters obtained from the calibration curve using the equations LOD = 3 *s_a_/b* and LOQ = 10 *s_a_/b* where *s_a_* is the standard deviation of the y-intercept of the regression line and *b* is the slope of the calibration curve [34]. The limit of detection for gallic acid in the form of dry residue on GCE for peak P1 was 5.13 μmol/L (0.87 mg/L) while the limit of quantification was 15.55 μmol/L (2.64 mg/L).

Square-wave voltammetry was also performed to determine the antioxidant capacity of the dry residue of extract from plant extracts on the surface of GCE immersed into a Britton-Robinson buffer solution (pH 2.5) for different medicinal plants as shown in Figure 6a—chamomile, Figure 6b—yarrow and Figure 6c—St. John’s wort. All three consist of only one oxidation peak at the potentials *E*_P_ = 1.071 V, *E*_P_ = 0.795 V, and *E*_P_ = 0.720 V vs. Ag/AgCl/3 mol/L KCl, respectively. The voltammogram of the dry residue of extract from the Curry plant is rather different with distinctive two oxidation peaks at the potentials *E*_P1_ = 0.491 V and *E*_P2_ = 0.927 V, and one poorly developed oxidation peak at *E*_P3_ = 0.674 V (Figure 6d). Additionally, a change in the pH values, ranging from 2.5 to 9.5, for all obtained extracts was evaluated to assess the electrochemical mechanism of the obtained extracts and gain insight into their antioxidant activity. The pH of the solution affects the voltammetric response of each sample, i.e., the oxidation current is strongly dependent on pH (data not shown). The potential of the peak for both yarrow and St. John’s wort extracts is a linear function of pH over the pH range from 2.5 to 9.5, and can be expressed by the equations: *E*_P1_ (V) = −0.047 pH + 0.901, *r* = 0.980, and *E*_P1_ (V) = −0.048 pH + 0.818, *r* = 0.991, respectively. Potentials for peak P1 and P3 of curry plant where linearly dependent over the tested pH range and are described by equations: *E*_P1_ (V) = −0.032 pH + 0.565, *r* = 0.979 and *E*_P3_ (V) = −0.023 pH + 0.987, *r* = 0.962, respectively, while responses for second poorly developed peak P2 of curry plant and for chamomile extract were pH-independent.

Under acidic experimental conditions (pH 2.5), square-wave voltammograms were recorded by changing the frequency in the range from 10 to 200 Hz for all tested samples. The first peak potential was independent of the logarithm of frequency (10 Hz < *f* < 200 Hz) for both St. John’s wort and the curry plant extracts. However, the plot of the net peak potential for oxidation peak P3 of curry plant, but also chamomile and yarrow versus the logarithm of SWV frequency is characterized by linear line with different slopes as follows: 0.886, 1.033, and 0.725 V/d.u., respectively. 

Cyclic voltammograms of plant extracts were recorded in Britton-Robinson buffer (pH 2.5) within the potential range from 0.2 to 1.6 V and at a scan rate of 25 mV/s as a support for the electrochemical mechanism of extracts (data not shown).

### 2.5. Determination of Antioxidant Capacity (AOC)

To assess antioxidant activity using voltammetric analysis, the area under the curve (AUC) was integrated, the value of which represents an estimate of the total antioxidant activity of the extracts. To express total antioxidant capacity in gallic acid equivalents (GAE), a calibration was carried out in which AUC was plotted against different concentrations of gallic acid standard. Additionally, two other spectrophotometric methods regarding the assessment of antioxidant activity were employed, namely DPPH and FRAP assay. As can be seen in Table 4, the highest antioxidant activity based on the AUC value was obtained for chamomile extract, followed by curry plant > yarrow > St. John’s wort extracts. However, compared to chamomile antioxidant activity, curry plant extract show 1.6 folds lower activity (*p* < 0.001), while around 6 folds lower activity (*p* < 0.001) was obtained for yarrow and St. John’s wort extracts. No significant difference in antioxidant activity was observed between yarrow and St. John’s wort extracts by implementing voltammetric analysis. Results of the DPPH assay, based on spectrophotometric measurements, also revealed the highest antioxidant activity for chamomile extract (25 mg/mL) with the inhibition percentage around 90%, followed by curry plant extract with an inhibition percentage around 50%, which is a significant decrease in activity (*p* < 0.001). As depicted in Table 4, using this method, the results were obtained for all tested samples mostly following the same order of activity as for voltammetric analysis: chamomile > curry plant > St. John’s wort > yarrow. 

Antioxidant activity in the CO_2_ supercritical fluid extracts varied from 7.3 to 65.7 mM GAE/g extract for the FRAP assay (Table 4). Samples with a relatively high FRAP activity were St. John’s wort and yarrow, although the latter showed two-folds (*p* < 0.001) lower activity. Chamomile and curry plant extracts showed similar antioxidant activity, but significantly lower when compared to the other two samples (*p* < 0.001).

## 3. Discussion

Supercritical fluid extraction as a sustainable green technology has led to a wide range of applications since the past decade. For each type of extract, various sets of parameters are important. The most important variables in the case of extraction of bioactive compounds are temperature, pressure, and static extraction time [35]. Kotnik et al. [36] investigated the supercritical CO_2_ extraction of chamomile flower heads and obtained the yield of 3.81%, which is lower than the obtained yield in our study (Table 1). Additionally, Molnar et al., 2017 also evaluated the usage of SFE to obtain high-value bioactive compounds in *Matricaria chamomilla* with obtained extraction yield of 3.64%. Furthermore, they obtained a higher yield of essential oil with supercritical CO_2_, since the degradation of thermolabile compounds (e.g., matricine) is minimized. The extraction yield of the Curry plant is in accordance with the literature data [37]. Smaller discrepancies in extraction yield can be explained by the different growth locations of plants and extraction parameters. It also needs to be stressed that is common that some very precious compounds are extracted in low yields and the process of isolation in such cases is completely different. Therefore, the SFE technique has shown to be an attractive alternative to the other currently used conventional methods in the point of obtaining plant extracts with diverse chemical compositions while complying with the green chemistry principles. However, as seen in Figure 2, experimental design has high influence on the extraction, so for the extraction of targeting substances more experiments are needed, especially in a region where pressure and temperature will vary in the vicinity of the near-critical region (NCR).

There were found several peculiarities among detected classes of compounds and their distributions in the analyzed extracts. Chamomile and yarrow were the only two extracts containing fatty acids and their derivatives, flavonoids, and coumarins. The same plant extracts had the highest number of detected terpenoids and sesquiterpenes. Compounds detected for chamomile were in accordance with already published research [38]. The highest number of functionalized phytosterols were found in curry plant and St. John’s wort extracts. Curry plant extract at the same time was found to be most rich with different polyphenols. Compounds from the PPAP family were found only in St. John’s wort extract (Appendix A).

Supercritical CO_2_ extraction enables a high yield of different bioactive compounds from the sample. In this research, naturally occurring organic compounds are of interest, and since they could act as antioxidants through electron transfer, it was considered that they would be largely responsible for the antioxidant activity of analyzed samples. It is known that the oxidation potentials of flavonoids correlate with their antioxidant activity, i.e., flavonoids with less positive oxidation potential, possess higher radical scavenging activity [39,40]. The extracts have shown a strong difference concerning total phenol content (going from 0.8 mg GAE/g extracts in the case of Yarrow to 5.6 mg GAE/g in the case of Curry plant extract). Extracts of Chamomile and St. John’s wort express quite similar total phenol content (2.3 and 2.7 mg GAE/g extract). The results obtained for supercritical CO_2_ extracts show that relatively low polyphenolic content was observed in all tested samples, which is in accordance with the results obtained using LC-HRMS analysis. This is also in agreement with literature data [41] were SCF analysis of plants from the Asteraceae family, among which was yarrow, revealed around 16-fold lower total phenolic content when compared to ultrasound-assisted extraction. For comparison, this content is even lower in our study, around 200-fold lower. However, this is not surprising since SC-CO_2_ is best suitable for extracting oils and lipophilic compounds [42,43]. Although the same trend was found for ethanolic extracts from curry plant, chamomile and yarrow, i.e., the highest TP content was found in curry plant [44], the overall phenolic content was much higher, 800-fold higher for ethanolic extracts from chamomile and yarrow, while more than 1500-fold higher for ethanolic extracts from curry plant. Nevertheless, the highest TP content was found in curry plant extract, which correlates with detected compounds (see Appendix A).

Since voltammetric methods determine antioxidant activity by measuring the electron-donating capacity, these methods can be used for evaluating antioxidants in obtained extracts. Similar to flavonoids, in phenolic acids, stronger antioxidant capacity is achieved by increasing the number of -OH groups attached to the aromatic ring [45,46]. For that reason, gallic acid was used as a polyphenolic standard and analyzed by square-wave voltammetry (SWV). Prior to the determination of antioxidant activity, sample extracts along with gallic acid were analyzed in detail by square-wave voltammetry on a glassy carbon electrode (GCE). Since changing the frequency (10–200 Hz) didn’t influence the peak potential and the logarithm of frequency is linear with the slope 0.349 V/d.u, irreversibility of the oxidation process of gallic acid was confirmed, i.e., oxidation of a galloyl moiety in the aromatic structure. This is in accordance with other published data [47,48] with slight differences that could be ascribed to different electrodes and electrolytes used in the experiments. Determination of recovery for peak P1 showed the accuracy of the implemented method with low standard deviations for both current and potential between measurements, which implies a non-contaminated surface of the working electrode and high repeatability in the identification of this oxidation peak. Linearity was obtained for a relatively wide range of concentrations (1–20.0 μM), while both calibration lines (using peak height and area) indicated low LOD and LOQ values which is in agreement with other literature data [47]. The results obtained for gallic acid in a form of dry residue in Britton Robinson buffer (pH 2.5) are good and acceptable. Reversibility of reactions for St. John’s wort and curry plant extracts was confirmed by changing the frequency and plotting the first peak P1 potential with the logarithm of frequency (10 Hz < *f* < 200 Hz). However, the linear dependence of oxidation peak potentials of P3 for curry plant, but also P1 for chamomile and yarrow extracts with the logarithm of frequency confirms that these processes are irreversible [49]. Difference in an (ir)reversibility for peak P1 and P3 in curry plant indicate that more than one electroactive molecule has reacted with electrode surface, probably belonging to the group of functionalized polyphenols but with different affinity to release an electron (i.e., different peak potentials). Subsequently, by conducting the cyclic voltammetry (CV) in Britton-Robinson buffer (pH 2.5) all plant extracts showed lower sensitivity but confirmed the (ir)reversibility of these processes for each obtained peak. Because SWV yielded more and better-defined peaks than the CV, it was used for the determination of the antioxidant profile of medicinal plant extracts.

Since antioxidants act by several mechanisms and a single assay cannot accurately reflect all of the antioxidants in a complex fraction [50,51], and to evaluate voltammetric method, in this research antioxidant activity was also measured by employing additional two spectrophotometric methods. The ferric reducing antioxidant power (FRAP) mechanism measures the antioxidant action via single electron transfer (a SET mechanism) rather than detecting compounds that act only by radical quenching (i.e., hydrogen atom transfer (HAT)) [52] so it can be used as a good method to estimate the mechanism of antioxidant activity, while DPPH assay relies on both mechanisms (SET and HAT) and is most commonly used method for estimation of total antioxidant activity. DPPH activity is relatively low when expressed per gram of extract for all tested samples, which could be explained by low total polyphenolic content since it is known that polyphenols exhibit higher antioxidant activity [53]. However, some other phenolic compounds such as flavonoids and coumarins, together with prenylated phloroglucinols also exhibit antioxidant activity [22]. Chamomile extract showed the highest antioxidant activity, followed by curry extract in which polyphenols contribute to the antioxidant activity. Results of the DPPH assay are in agreement with other studies on supercritical CO_2_ extracts from these medicinal plants. When analyzing and correlating different extraction methods, Molnar et al. [38] found that the supercritical CO_2_ extracts along with hexane extracts did not show any significant activity, explained by the lower yield of polyphenols using this extraction method. Additionally, García-Risco et al. [41] evaluated the antioxidant activities of four different plants, among which was yarrow, and also observed lower antioxidant activity in the extract obtained by supercritical CO_2_ extraction when compared to conventional methods. Results obtained using FRAP assay differ from the results obtained using DPPH assay, thus indicating that the chemical composition of obtained extracts is significant to the mode of action. The latest study shows that phloroglucinols exhibit different modes of action [54], similar to the polyphenols [55]. Also, SWV revealed that most of the area that is quantified from the voltammograms has a potential between 0.600 V and 0.900 V (Figure 6). The FRAP assay quantifies antioxidant compounds that have redox potentials below 0.700 V (i.e., the redox potential of the Fe^3+^−TPTZ complex) [52] which corresponds to the observed highest activity for St. John’s wort extract. To correlate total phenolic content (TPC) with antioxidant activity (SWV, DPPH, and FRAP assays), Pearson’s correlation coefficient [56] was applied. Also, this correlation was evaluated between SWV analysis and spectrophotometric methods. The TPC showed a higher negative correlation to DPPH with Pearson’s correlation coefficient of −0.639, while the medium negative correlation between TPC versus FRAP and SWV was found with Pearson’s correlation coefficients of −0.279 and −0.303, respectively. However, the significance level wasn’t sufficient enough suggesting that the antioxidant activity of these medicinal plant extracts is dependent on not only phenols but also other compounds that exhibit different antioxidant modes of action. Multiple regression analysis between SWV and DPPH and FRAP showed a good correlation. The Pearson’s correlation coefficient for SWV versus DPPH was a positive value of 0.818, while for FRAP assay high negative value of −0.640 was obtained, both marginally significant (*p* = 0.07). This provides information that SWV can be used to assess the antioxidant activity of the samples. However, one should note that for positive voltammetric analysis, compounds found in samples must be electroactive to provide a signal. Since CO_2_ extracts have different chemical compositions, it is expected that not all molecules in the sample will be electroactive. Thus, Pearson’s correlation coefficients between SWV *versus* DPPH and FRAP show good results implying potential usage of voltammetry to quickly determine antioxidant activity.

## 4. Materials and Methods

### 4.1. Plant Extracts Preparation

Freshly picked herbal plants collected from the wider Primorska region, Slovenia were dried in the stream of dry air at 50 °C. The drying was stopped when the humidity of the sample was under 5 wt. %. Afterward, the samples were grinded and particles with 2 mm diameter were chosen for further analyses. The extraction of herbal plants (2 kg) was done by applying a BBES 2.0 extraction system (Waters, Milford, MA, USA) equipped with two 10 L extraction vessels and three consecutive connected 2 L collection vessels. Samples were extracted according to the internal extraction method to get the highest yield of bioactive component, as is listed in Table 5.

The mixture of supercritical fluid and extract was separated into three consecutive collection vessels connected in a row. The separation conditions are listed in Table 6.

Herbal extracts from collection vessels 2 and 3 were combined and mixed together, then stored at 4 °C for further analysis. The extraction yields were calculated based on the initial dried raw material mass. The extracts from collection vessel 1, containing waxes and heavily soluble lipophilic compounds, were discarded and were not analyzed.

### 4.2. Identification of the Organic Compounds

#### 4.2.1. Gas Chromatography-High-Resolution Mass Spectrometry (GC-HRMS Analysis)

GC-HRMS analysis was carried out using a Pegasus^®^ GC-HRT^+^ 4D high-resolution mass spectrometer (LECO Corporation, Saint Joseph, MI, USA) combined with an Agilent 7890A gas chromatograph (Agilent Technologies, Palo Alto, CA, USA) equipped with a LECO quad-jet cooled thermal modulator and a secondary column oven. Spectra collection, data processing, and general system control were conducted by means of ChromaTOF^®^ software (Version 5.20, LECO Corporation, Saint Joseph, MI, USA). Two-dimensional comprehensive gas chromatography (GC×GC) separation was carried out with an Rxi-5SilMS 30 m × 0.25 mm (id) × 0.25 µm (df) (Restek, Bellefonte, PA) as the first dimension column and an Rxi-17SilMS column 1 m × 0.25 mm (id) × 0.25 μm (df) (Restek, Bellefonte, PA, USA) for the second dimension column. The GC oven program was as follows: a 2 min isothermal hold at 40 °C, then ramping at 20 °C/min to 280 °C followed by a 10 min isothermal hold at 280 °C. The secondary oven temperature was set to 20 °C higher than the primary oven. The modulator temperature was offset by 15 °C above the secondary oven temperature and the modulation period was set to 6 s. Two hundred mass spectra per second (*m/z* 15–800) were acquired with the resolving power of 25,000 using electron ionization (EI) with 70 eV. Any additional conditions were taken as described previously [57,58].

#### 4.2.2. Liquid Chromatography High-Resolution Mass Spectrometry (LC-HRMS Analysis)

LC-HRMS analysis was performed using HPLC system LC-30 Nexera (Shimadzu, Kyoto, Japan) combined with quadrupole—time-of-flight (QTOF) mass spectrometer TripleTOF 5600+ (AB Sciex, Concord, Canada. Chromatographic separation was performed on a Nucleodur PFP column (Macherey-Nagel, Düren, Germany), 150 × 2 mm, 1.8 µm, packed with pentafluorophenyl stationary phase, in gradient mode. Eluent composition: deionized high-purity Milli-Q H_2_O (with 0.1% formic acid) and acetonitrile (with 0.1% formic acid), gradient program: 0–1 min 10% acetonitrile, 1–15 min increase in acetonitrile content up to 100%, 15–25 min 100% acetonitrile. Flow rate—0.25 mL/min, column temperature 40 °C, injection volume 5 µL. All solvents were taken HPLC grade.

Ion source parameters: Electrospray ionization in positive and negative mode (ESI+ and ESI-), Curtain gas (CUR)—30 psi, Nebulising and drying gases (GS1 and GS2)—40 psi, Temperature (TEM)—300 °C, Voltage (ISVF)—5500 V (−4500 V in Negative mode), Declustering potential (DP)—80 V. The non-targeted screening was performed in Information Dependent Acquisition (IDA) mode. Mass range in TOFMS mode (MS1): 100–1000 Da. Ions in MS1 spectra after HPLC separation with intensity greater than 100 cps were fragmented automatically. Collision-induced dissociation (CID) was performed using collision energy (CE) 40 eV with CE spread 20 eV. Mass range in Product ion mode (MS2): 20–1000 Da.

### 4.3. The Total Phenolic Content (TPC)

All spectrophotometric measurements were carried out using a UV/Vis microplate reader (Infinite M200 PRO, TECAN, Männedorf, Switzerland) in triplicates. The Folin-Ciocalteu method was used for total phenolic content (TPC) measurement with some adaptations [59]. Briefly, 100 μL extract was mixed with 750 μL of Folin–Ciocalteu reagent (previously diluted tenfold with distilled water; Kemika, Zagreb, Croatia) and allowed to stand at room temperature for 5 min. Next, 750 μL sodium bicarbonate (Kemika) solution (60 g/L) was added to the mixture. After incubation for 90 min at room temperature, the absorbance was measured at 750 nm. Total phenolics were calibrated against gallic acid (>97.5%, Sigma Aldrich, St Louis, MO, USA) standards and are expressed as mg gallic acid equivalent (GAE)/g extract.

### 4.4. The Radical Scavenging Assay (The DPPH Assay)

For the radical scavenging assay, the 1,1-diphenyl-2-picrylhidrazine (DPPH, ≥98%, Sigma Aldrich) radical was used [60] and ascorbic acid (≥99%, Sigma Aldrich) was the reference compound. For the DPPH assay, 250 μL of extract diluted in ethanol (p.a., Kemika) was mixed with 500 μL of deionized water and prepared DPPH reagent in methanol (72 μg/mL, p.a., Kemika). The reaction mixture was kept in the dark for 30 min, after which the absorbance of the solution was measured at 517 nm in a 96-well plate. Appropriate blanks (ethanol) and standards (ascorbic acid solutions in ethanol) were run simultaneously. Results were expressed as milligram ascorbic acid equivalents per gram of sample (mg AAE/g sample).

### 4.5. The Ferric Reducing Antioxidant Power (The FRAP Assay)

The FRAP assay was carried out according to the method of Benzie and Strain with minor modification [61]. In brief, FRAP reagent was freshly prepared by mixing equal volumes of a 10 mmol/L 2,4,6-tripyridyl-S-triazine (TPTZ, ≥98%, Sigma-Aldrich) solution in 40 mmol/L HCl (p.a., Kemika) and an aqueous 20 mmol/L FeCl_3_ (p.a., Kemika) solution and diluting this mixture five times in a 0.25 mol/L acetate buffer (pH 3.6), followed by warming to 37 °C. Next, 100 μL of sample extract or gallic acid as a positive control was mixed with 3.9 mL of FRAP reagent, and the absorbance of the reaction mixture was measured at 593 nm after incubation for 10 min at 37 °C. Results were calculated as mg FeSO_4_/g sample and then normalized to representative concentrations of gallic acid equivalents (mM GAE/g sample) for comparison purposes.

### 4.6. Voltammetric Analysis

The electrochemical experiments were carried out using the computer-controlled PalmSens electrochemical system (Houten, the Netherlands) with PSTrace software using a glassy carbon electrode (GC electrode, BASi, diameter 3 mm) as a working electrode, an Ag/AgCl (3 mol/L KCl) electrode as a reference electrode and a platinum wire as a counter electrode. All potentials were expressed versus Ag/AgCl (3 mol/L KCl) reference electrode. Cyclic (CV) and square-wave (SWV) voltammetry were performed on all samples in Britton-Robinson buffer as an electrolyte solution. The working electrode was cleaned by polishing with 3 μm alumina powder for 1 min and rinsed with ethanol between runs.

Analysis of standards and samples was performed in a way that the GC working electrode was dipped into the ethanol solutions of samples and left to dry at room temperature for a few minutes. By this procedure, the surfaces of the GC electrode became contaminated with dry residue of samples or standards. The working electrode was immersed in the electrolyte only during the voltammetric measurements. Less than 1 mm of the graphite rod was immersed in the electrolyte.

The solutions were degassed with high purity nitrogen for at least 20 min before all electrochemical measurements. A nitrogen blanket was maintained thereafter. Unless otherwise stated, all measurements were performed in triplicates. CV experiments on modified GC electrode were performed at a potential scan rate of 25 mV/s, while SWV was performed using a potential step increment of 2 mV and a square-wave amplitude of 50 mV. The frequency varied from 10 to 200 Hz. The change in pH value was also analyzed.

## 5. Statistical Analysis

The obtained values for evaluation of antioxidant activity are expressed as mean values with standard deviations of three replicates. The differences between the means were analyzed by Tukey’s test of One-Way ANOVA using GraphPad Prism 8.0 (GraphPad Software Inc., San Diego, CA, USA). Values of *p* < 0.05 and lower were considered as significantly different. Correlations and regression analysis describing the antioxidant activities of TCP, DPPH, and FRAP assay along with voltammetric analysis (SWV) were conducted using the regression program in GraphPad Prism 8.0.

## 6. Conclusions

In this study, we tested CO_2_ SFE efficiency of four model medicinal herbs in order to make the comparison in terms of extraction yields, phenolic profiles, and antioxidant potential among them and find the minimum of common parameters which would give satisfactory extraction yields of antioxidants with desired antioxidant capacity together with chemical analysis of most important compounds. For this reason GC-HRMS and LC-HRMS, spectrophotometric methods, as well as electrochemical methods, have been applied. The investigated extracts from medicinal plants had a diverse chemical composition, consequently leading to different observed antioxidant activities. Overall results have shown that the same method could be used for the routine extraction of various plants for common purposes (like food supplements). In the case where compounds of special properties and interest are investigated, the modifications of methods and development of specific protocols are needed. The experimental design with defined conditions is very important for conducting SFE successfully for each unique matrix.

## Figures and Tables

**Figure 1 molecules-26-05697-f001:**
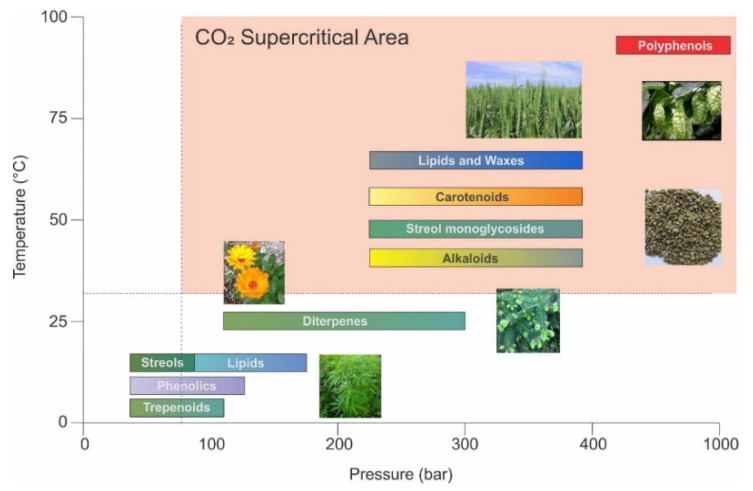
Schematic diagram showing control parameters of Supercritical Extraction (SCE) processes for the extraction of bioactive compounds.

**Figure 2 molecules-26-05697-f002:**
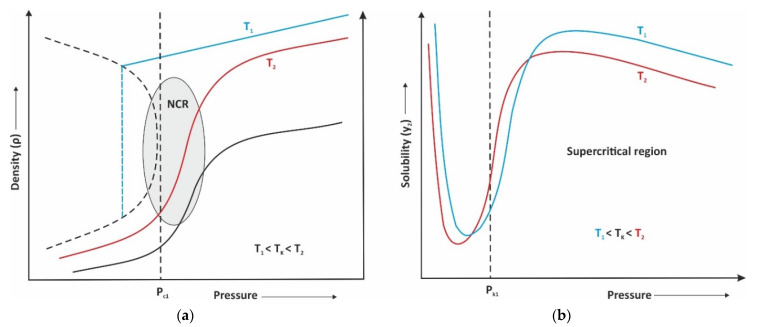
The interconnection between the (**a**) density and (**b**) solubility by increasing the pressure.

**Figure 3 molecules-26-05697-f003:**
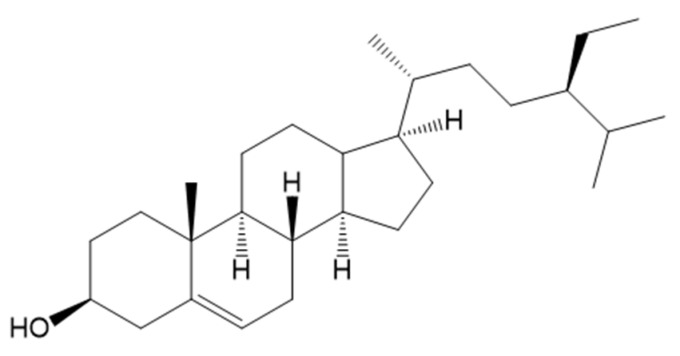
Skeletal structure of β-sitosterol—an example of a phytosterol.

**Figure 4 molecules-26-05697-f004:**
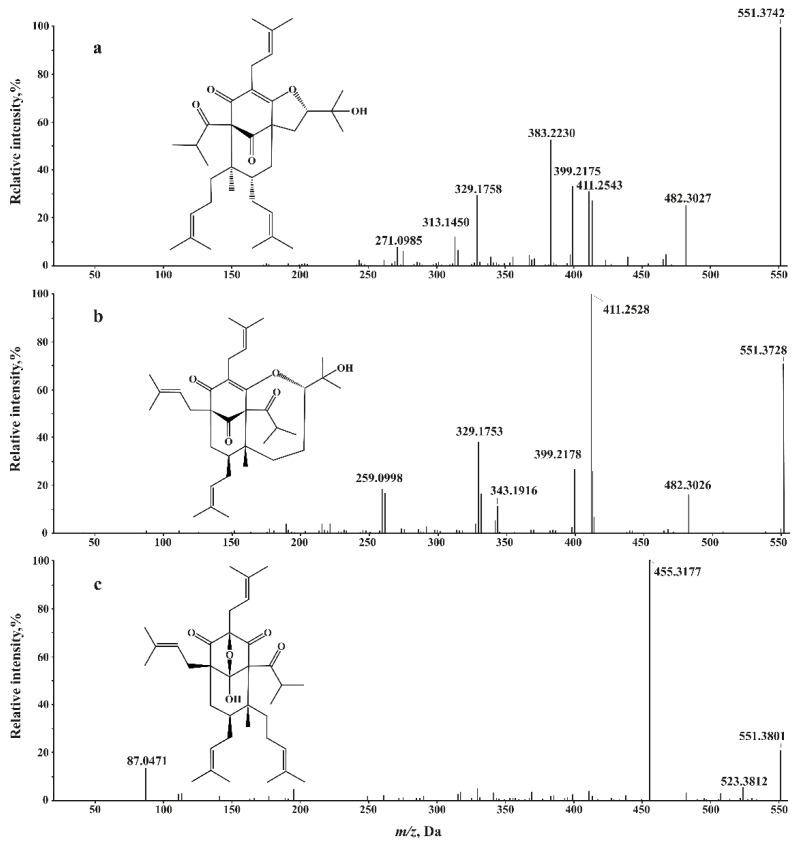
ESI(-) CID mass spectra of St. John’s wort components at (**a**) RT 12.61 min, (**b**) 13.65, and (**c**) 14.7 min.

**Figure 5 molecules-26-05697-f005:**
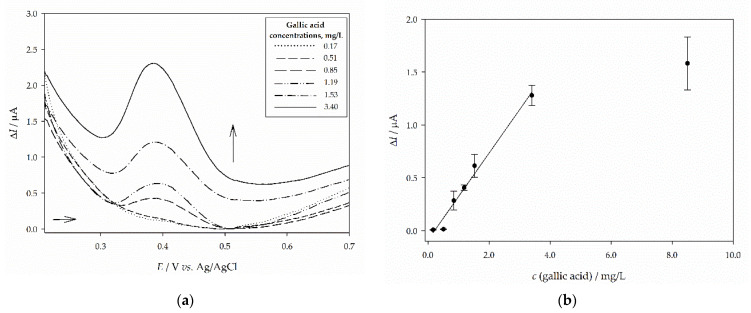
Experimental analysis of polyphenolic standard gallic acid: (**a**) square-wave voltammograms (SWV) obtained on the surface of glassy carbon electrode (GCE) and immersed in Britton Robinson buffer (pH 2.5) for concentration range 0.17–3.40 mg/L; (**b**) dependence of the net peak currents on the concentration of gallic acid with the linear regression line. The frequency is 10 Hz, pulse amplitude is 50 mV, and the step potential is 2 mV.

**Figure 6 molecules-26-05697-f006:**
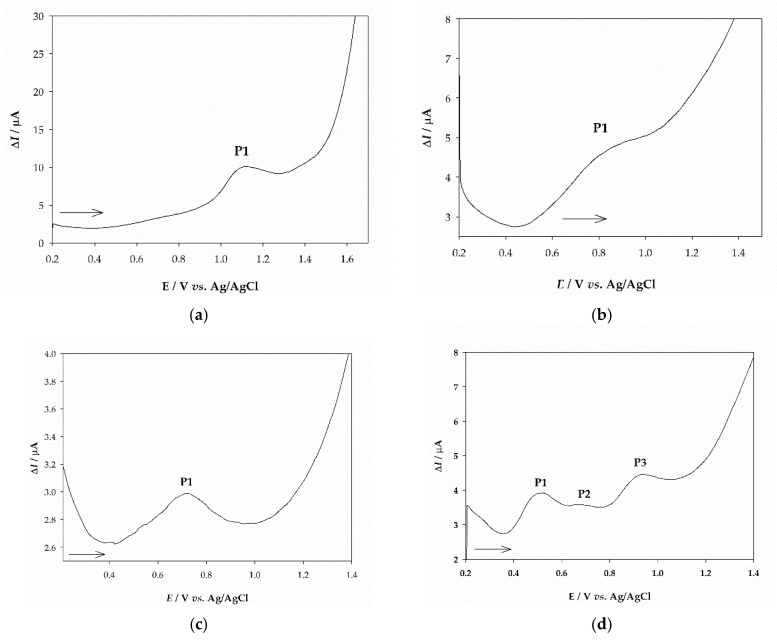
Square-wave voltammograms (SWV) of supercritical CO_2_ extracts obtained from four medicinal plants: (**a**) Chamomile; (**b**) Yarrow; (**c**) St. John’s wort; (**d**) Curry plant in the form of microfilm immobilized on the surface of glassy carbon electrode (GCE) and immersed in Britton Robinson buffer (pH 2.5). The experimental conditions are as depicted in Figure 5.

**Table 1 molecules-26-05697-t001:** Plant extract yields obtained with supercritical CO_2_ extraction.

Plant	Yield, %			
	Collection Vessel 1	Collection Vessel 2	Collection Vessel 3	Total
Chamomile	2.20	3.20	1.80	7.20
St. John’s wort	1.74	2.20	0.98	4.92
Curry plant	1.25	1.64	0.86	3.75
Yarrow	1.63	1.85	0.92	4.40

**Table 2 molecules-26-05697-t002:** List of organic compounds identified in plant extracts with GC×GC-HRMS.

No.	Formula	Compound	RT ^1^	Chamomile	Yarrow	St. John’s wort	Curry Plant
1	C_15_H_24_	Sesquiterpene	1072 s, 2.450 s			+	
2	C_15_H_24_	*cis*-β-Farnesene	1088 s, 2.325 s	+			
3	C_15_H_24_	Sesquiterpene	1112 s, 2.450 s			+	
4	C_15_H_22_	Sesquiterpene	1144 s, 2.550 s			+	
5	C_15_H_24_O	Spathulenol	1184 s, 2.625 s	+			
6	C_15_H_24_	Sesquiterpene	1192 s, 2.625 s			+	
7	C_12_H_18_O	C12H18O	1208 s, 2.632 s	+			
8	C_15_H_24_O	Sesquiterpenoid	1208 s, 2.600 s		+	+	
9	C_15_H_24_	Sesquiterpene	1224 s, 2.600 s		+	+	
10	C_15_H_26_O_2_	(-)-Bisabolol oxide B	1232 s, 2.575 s	+			
11	C_15_H_24_O_2_	Bisabolone oxide A	1256 s, 2.650 s	+			
12	C_15_H_26_O_2_	(-)-α-Bisabolol oxide A	1296 s, 2.675 s	+			
13	C_13_H_12_O_2_	(*E*)-2-(Hexa-2,4-diyn-1-ylidene)-1,6-dioxaspiro[4.4]non-3-ene	1376 s, 3.100 s	+			
14	C_13_H_12_O_2_	(*Z*)-2-(Hexa-2,4-diyn-1-ylidene)-1,6-dioxaspiro[4.4]non-3-ene	1384 s, 3.350 s	+			
15	C_17_H_20_O_5_	Matricarin	1664 s, 3.250 s	+			

^1^ RT represents the retention time.

**Table 3 molecules-26-05697-t003:** Total phenolic content (TPC) of sample extracts (mean ± SD; *n* = 3) expressed as mg GAE (Gallic Acid Equivalents) per g of extract.

Sample Extract	TPC Content (as Gallic Acid Equivalents/g Extract)
Chamomile	2.30 ± 0.02 mg/g ^abc^
St. John’s wort	2.70 ± 0.02 mg/g ^ade^
Curry plant	5.60 ± 0.03 mg/g ^bdf^
Yarrow	0.80 ± 0.02 mg/g ^cef^

Values sharing common letters within the same column indicate a statistically significant difference at a 0.001 probability level.

**Table 4 molecules-26-05697-t004:** Antioxidant activities in extracts obtained using supercritical CO_2_ extraction from four medicinal plants measured by three antioxidant assays (SWV, DPPH, and FRAP). Results are presented as mean value ± standard deviation (*n* = 3).

Sample Extract	SWV mg GAE ^1^/g Extract	DPP Hmg AAE ^2^/g Extract	FRAP mM GAE ^1^/g Extract
Chamomile	0.265 ± 0.0 ^abc^	47.3 ± 0.6 ^abc^	7.3 ± 0.3 ^ab^
St. John’s Wort	0.037 ± 0.0 ^ad^	21.0 ± 0.8 ^ade^	65.7 ± 0.8 ^acd^
Curry plant	0.164 ± 0.0 ^cde^	25.7 ± 1.3 ^cef^	10.5 ± 0.5 ^de^
Yarrow	0.050 ± 0.0 ^be^	5.7 ± 0.9 ^bdf^	29.3 ± 0.5 ^bce^

^1^ Gallic acid equivalents. ^2^ Ascorbic acid equivalents. Values sharing common letters within the same column indicate a statistically significant difference at a 0.05 probability level.

**Table 5 molecules-26-05697-t005:** The extraction parameters.

Plant	T [°C]	P [bar]	ρ [kg/m^3^]	Θ [g/min]	T [min]	S/S ^1^
Chamomile	48	200	795.5	120	900	54
St. John’s wort	40	230	863.7	120	600	36
Curry plant	40	230	863.7	120	990	60
Yarrow	40	200	839.9	120	900	26

^1^ S/S ratio represents the amount of fresh solvent applied to raw material.

**Table 6 molecules-26-05697-t006:** Separation conditions used in vessels.

	T [°C]	P [bar]
Collection vessel 1	45	150
Collection vessel 2	40	75
Collection vessel 3	30	50

## Data Availability

The data presented in this study are available for limited time on request from the corresponding author.

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
