# Peer review of "Supercritical Carbon Dioxide Extraction of Four Medicinal Mediterranean Plants: Investigation of Chemical Composition and Antioxidant Activity"

_molecules, 2021, doi:10.3390/molecules26185697_

Round 1

Author Response

Answers to Reviewer 1

Proposed corrections to MS.

  1. The aim of this work was non-target analysis with tentative identification of all possible components. The results obtained for supercritical CO2 extracts showing a relatively low polyphenolic content in all tested samples, were referred to the fact that CO2 SFE is best suitable for extracting oils and lipophilic compounds. However, the trend found for, e.g., ethanolic extracts from the plants under study related to any quantitative data remains non-discussed. It would be very useful to evaluate ratios, at least for few compounds, between the polyphenols level obtained with solvent (according to a literature data) and with SC-CO2. Low total polyphenolic content results in lower antioxidant activity (as authors point out themselves at Line 431). Thus, the addition of the said quantitative data would help readers to judge on the framework of applicability of CO2 SFE to gain in polyphenols.

We thank the reviewer for the suggestion. Since quantification of each polyphenol was not the aim of this study, we can only discuss total phenolic content in terms of ratios with other conventional extraction methods. As suggested, we modified that part of the discussion and added ratios (please see page 11).

  1. Line 64: "widely used as SFC ...", I suppose, SFC should be replaced by SCF, since this abbreviation is given at Line 61, "Supercritical fluids (SCF)".

We thank the reviewer for observing this mistake. It was corrected.

  1. Line 121. During the identification of organic compounds in plant extracts with GC-HRMS, was any quantitative characterization performed? E.g. peak square (relative) values, as usual, offer a rather good visualization of the whole picture, therefore being within the intent scope of MS, without the use of any standards, on the basis of NIST14 EI spectral libraries (Line 124).

No, in this research we didn't do any quantitation, because of the lack of standards. The main goal was to evaluate supercritical fluid extraction and qualitatively identify non-targeted compounds.

  1. Line 332: "... depicted in Table 6". I suppose it should be replaced by "... depicted in Table 4", since Table 6 (Line 483) reflects "separation conditions used in vessels".

We thank the reviewer for observing this mistake. The change is made and now states: “..depicted in Table 4”.

  1. Line 335, and Lines 546-555. In the FRAP assay, Trolox solution was used in the procedures of mixing with FRAP reagent and measuring the absorbance of this reaction mixture. But results "were normalized to representative concentrations of gallic acid equivalents (mM GAE/ g sample)". Why not use Trolox equivalents for comparison purposes? If not, any explanations related to the implementation of measured mixture (with Trolox) distinct from the final antioxidant equivalent (GA) should be done, may be, a kind of recalculation performed, if any.

We thank the reviewer for observing this mistake. Namely, Trolox was not used as it is an analog of vitamin E, and in this paper the aim represents polyphenols, so accordingly gallic acid was used to normalize the results so that they can be comparable to the other used methods. Also, the same thing was performed for voltametric measurements. To make additional clarification, FRAP is measured according to the standard, FeSO4 and was first expressed as Ferrous equivalents (mmol/g). Additionally, gallic acid was also measured as samples, and samples were then normalized and expressed as Gallic acid equivalents. To make sure that the used protocol is completely understandable to the reader, the changes were made in Section 4.5.

  1. Line 476. "and 3 consecutive ... collection vessels." should be replaced by " and 3 consecutive ... collection vessels (CV1, CV2, CV3)." Provided that the paper is read successively, i.e. section "2.1. Plant extract yields" is followed by "4.1. Plant extracts preparation", so in 2.2, Table 1, the abbreviations CV1, CV2, CV3 are not all apparent at first glance. I realize that "CV" is for cyclic voltammetry, but the latter abbreviation is without 1, or 2, or 3.

We thank the reviewer for his comment and suggestions. To avoid misunderstanding or mixing CV (collection vessel) and CV (cyclic voltammetry), we decided to remove the abbreviation CV from Table 1 and write the whole name.

Reviewer 2 Report

Corrections

Table 2 – Add in legend what is 1072 s, 2.450 s

Results shown in Figure 3 - Fig. 3c mass spectrum contains an abundant peak at m/z 455.3177 is (M – H – 96)-, which did not correspond to the loss of CO molecule (The loss of CO correspond to 28 Da and the loss of CO2 correspond to 44 Da). Figure 3a mass spectrum exhibits the loss of 2 prenyl (C5H9) radicals resulting in m/z 413.2350 together with formation of m/z 383.2230 (C24H31O4) that may occur through the loss of CO in the fragment ion m/z 411.2543 (411.2543 - 383.2230 = 28). Figure 3b mass spectrum contains an abundant peak at m/z 411 corresponding to the loss of C5H10. Correct the proposed structures.

There were found several peculiarities among detected classes of compounds and their distributions in the analyzed extracts. Chamomile and Yarrow were the only two extracts containing fatty acids and their derivatives, flavonoids, and coumarins. The same plant extracts had the highest number of detected terpenoids and sesquiterpenes. This is in accordance with already published research [Molnar, M., Mendešević, N., Šubarić, D., Banjari, I., Jokić, S. Comparison of various techniques for the extraction of umbelliferone and herniarin in Matricaria chamomilla processing fractions. Chem. Cent. J. 2017, 11, 78. 705. It is only for Chamomile. The highest number of functionalized phytosterols were found in Curry plant and St. John’s Wort extracts. Curry plant extract at the same time was found to be most rich with different polyphenols. Compounds from the PPAP family were found only in St. John’s Wort extract. Cite table

The list of detected compounds (Cite Tables) was classified into several groups of naturally occurring organic substances: fatty acids and their derivatives, flavonoids and coumarins, functionalized phytosterols, polyphenols, sesquiterpenes and terpenoids, and prenylated phloroglucinols. If the molecular formula obtained by accurate mass measurement with ESI-HRMS coincided with that mentioned in the literature and the tandem mass spectrum contained fragment ions in accordance with the proposed structure, the assignment was done the same as in literature. However, many substances were impossible to clearly identify since there were several isomers present with the same molecular formulae and similar fragmentation patterns (the fragmentation pattern must be added in Table S1 and S2) in one sample. Such compounds were just assigned with a class. The full list of detected compounds by means of LC-MS is summarized in Table S1 and S2

Reversibility of reactions for St. John’s Wort and Curry plant extracts was confirmed by changing the frequency and plotting the first peak potential with the logarithm of frequency (10 Hz < f < 200 Hz). However, the linear dependence of oxidation peak potentials of P3 for Curry plant, but also Chamomile and Yarrow extracts with the logarithm of frequency confirms that these processes are irreversible???. By conducting the cyclic voltammetry in Britton-Robinson buffer (pH 2.5) all plant extracts showed lower sensitivity but confirmed the (ir)reversibility??? It is right?? of these processes. SWV yields more and better-defined peaks so it was used for the determination of the antioxidant profile of medicinal plant extracts. Explain better

Since voltammetric methods determine antioxidant activity by measuring the electron-donating capacity, these methods can be used for evaluating antioxidants in obtained extracts. Similar to flavonoids, in phenolic acids, stronger antioxidant capacity is achieved by increasing the number of -OH groups attached to the aromatic ring (see reference -  Brazilian Journal of Pharmacognosy 23(3): 542-558, May/Jun. 2013)

SC- CO2 is best suitable for extracting oils and lipophilic compounds.

Add the reference - wineza PA, Waśkiewicz A. Recent Advances in Supercritical Fluid Extraction of Natural Bioactive Compounds from Natural Plant Materials. Molecules. 2020;25(17):3847. Published 2020 Aug 24. doi:10.3390/molecules25173847

(The fragmentation pattern must be added in Table S1 and S2)

The presentation of the results in the Supplementary Tables is bad. I think it is important to include the results displayed in the mass spectra, as new compounds may have been extracted.

C19H30O10 - Smth glucosilated???? – What is this in Table S2

Table S1 and S2, what is RDB

Which is the difference among Table 2 and Table S1

Hydroxycynnamic acid glucoside? Correct - Hydroxycinnamic acid glucoside

Butyl cynnamate – correct Butyl cinnamate

Author Response

  1. Table 2 – Add in legend what is 1072 s, 2.450 s

RT stands for retention time, so these are first and second retention times and a legend was added in the footnote of Table 2.

  1. Results shown in Figure 3 - Fig. 3c mass spectrum contains an abundant peak at m/z 455.3177 is (M – H – 96)-, which did not correspond to the loss of CO molecule (The loss of CO correspond to 28 Da and the loss of CO2 correspond to 44 Da).

We thank the reviewer for the comment. This part was corrected to: “As it clearly comes from Fig. 4 only mass spectrum c contains a peak at m/z 523.3812 corresponding to loss of CO molecule, while the most abundant peak at m/z 455.3177 corresponds to the consequent loss of C5H8 fragment.

  1. Figure 3a mass spectrum exhibits the loss of 2 prenyl (C5H9) radicals resulting in m/z 413.2350 together with formation of m/z 383.2230 (C24H31O4) that may occur through the loss of CO in the fragment ion m/z 411.2543 (411.2543 - 383.2230 = 28).

The accurate mass of m/z 411.2543 corresponds to C26H35O4, so m/z 383.2230 (C24H31O4) results from loss of C2H4.

  1. Figure 3b mass spectrum contains an abundant peak at m/z 411 corresponding to the loss of C5H10. Correct the proposed structures.

We agree with the reviewer's comment. This moment was added in the text (please see page 6).

  1. There were found several peculiarities among detected classes of compounds and their distributions in the analyzed extracts. Chamomile and Yarrow were the only two extracts containing fatty acids and their derivatives, flavonoids, and coumarins. The same plant extracts had the highest number of detected terpenoids and sesquiterpenes. This is in accordance with already published research [Molnar, M., Mendešević, N., Šubarić, D., Banjari, I., Jokić, S. Comparison of various techniques for the extraction of umbelliferone and herniarin in Matricaria chamomilla processing fractions. Chem. Cent. J. 2017, 11, 78. 705. It is only for Chamomile.

This part was corrected to “Compounds detected for Chamomile were in accordance with already published research [37]”.

  1. The highest number of functionalized phytosterols were found in Curry plant and St. John’s Wort extracts. Curry plant extract at the same time was found to be most rich with different polyphenols. Compounds from the PPAP family were found only in St. John’s Wort extract. Cite table

We thank the reviewer for the suggestion, we added the citing Table in the text.

  1. The list of detected compounds (Cite Tables) was classified into several groups of naturally occurring organic substances: fatty acids and their derivatives, flavonoids and coumarins, functionalized phytosterols, polyphenols, sesquiterpenes and terpenoids, and prenylated phloroglucinols. If the molecular formula obtained by accurate mass measurement with ESI-HRMS coincided with that mentioned in the literature and the tandem mass spectrum contained fragment ions in accordance with the proposed structure, the assignment was done the same as in literature. However, many substances were impossible to clearly identify since there were several isomers present with the same molecular formulae and similar fragmentation patterns (the fragmentation pattern must be added in Table S1 and S2) in one sample. Such compounds were just assigned with a class. The full list of detected compounds by means of LC-MS is summarized in Table S1 and S2

We are having trouble understanding this comment. The aim of this study was to characterize the most abundant compounds present in SCF extracts from 4 medicinal plants, compare them and their antioxidant activity. The present molecules in samples are known from the literature, so we think that their assignment is on point. In Tables S1 and S2, we showed the structures that are most probable. We believe that the data is prepared in a way that is informative and good for this kind of study.

  1. Reversibility of reactions for St. John’s Wort and Curry plant extracts was confirmed by changing the frequency and plotting the first peak potential with the logarithm of frequency (10 Hz < f < 200 Hz). However, the linear dependence of oxidation peak potentials of P3 for Curry plant, but also Chamomile and Yarrow extracts with the logarithm of frequency confirms that these processes are irreversible???. By conducting the cyclic voltammetry in Britton-Robinson buffer (pH 2.5) all plant extracts showed lower sensitivity but confirmed the (ir)reversibility??? It is right?? of these processes. SWV yields more and better-defined peaks so it was used for the determination of the antioxidant profile of medicinal plant extracts. Explain better.

We thank the reviewer for the comment and suggestion. To clarify the processes, the discussion part was modified in a manner that each peak process is now clearly marked (please see page 12). The Curry plant showed more oxidation peaks, while 3 other plants showed only one. Because of that, a reader can be confused by (ir)reversibility of each process. Accordingly, we changed Figure 5, i.e. every peak is marked on the voltammogram so that reader can better follow the discussion. Regarding the SWV and CV, the text was also modified in a way that is now clear that CV is used a confirmative voltametric method to SWV. Some researchers use the only CV for measurement of antioxidant activity, however, higher sensitivity can be obtained by employing SWV. CV was in this case only supportive method to confirm the irreversibility or reversibility of each process for each plant extract. Hopefully, the made changes make the discussion more understandable.

  1. Since voltametric methods determine antioxidant activity by measuring the electron-donating capacity, these methods can be used for evaluating antioxidants in obtained extracts. Similar to flavonoids, in phenolic acids, stronger antioxidant capacity is achieved by increasing the number of -OH groups attached to the aromatic ring (see reference - Brazilian Journal of Pharmacognosy 23(3): 542-558, May/Jun. 2013)

We thank the reviewer for the suggestion. After reading the manuscript, we added the reference.

  1. SC- CO2 is best suitable for extracting oils and lipophilic compounds. Add the reference - wineza PA, Waśkiewicz A. Recent Advances in Supercritical Fluid Extraction of Natural Bioactive Compounds from Natural Plant Materials. Molecules. 2020;25(17):3847. Published 2020 Aug 24. doi:10.3390/molecules25173847

We thank the reviewer for the suggestion. The reference was added.

  1. (The fragmentation pattern must be added in Table S1 and S2)

Inclusion of mass spectra for all detected compounds would exceed not just normal number of pages, but even the size of the document because that is a very large amount of data. From that reason we decided not to add them.

  1. The presentation of the results in the Supplementary Tables is bad. I think it is important to include the results displayed in the mass spectra, as new compounds may have been extracted.

In our opinion, the data can be represented in different ways. We decided to show the results in a tables to provide a more informative overview. Classification of compounds into several groups of naturally occurring organic substances makes an easier view of what group of compounds is most represented in each plant extract.

  1. C19H30O10 - Smth glucosilated???? – What is this in Table S2

We apologize for this oversight. The wrong name is removed from the table.

  1. Table S1 and S2, what is RDB

RDB stays for rings and double bonds equivalent. For better understanding, we added the abbreviation and its meaning in the caption of the Table.

  1. Which is the difference among Table 2 and Table S1

We apologize for this oversight and thank the Reviewer for this observation. Those are the same tables. We removed Table S1 from the Supplementary data and consequently changed the numbering.

  1. Hydroxycynnamic acid glucoside? Correct - Hydroxycinnamic acid glucoside

We thank the reviewer. The mistake was corrected.

  1. Butyl cynnamate – correct Butyl cinnamate

We thank the reviewer. The mistake was corrected.

Reviewer 3 Report

The manuscript “Supercritical carbon dioxide extraction of four medicinal Mediterranean plants: Investigation of chemical composition and antioxidant activity” deals with the extraction by supercritical CO2 of active compounds from Chamomile, St. John’s Wort, Yarrow, and Curry plant. In particular, the chemical composition, phenolic content, and antioxidant activity of the extracts were analyzed to validate the process. The work is well organized and written; moreover, intriguing results were obtained. The publication is recommended; but, after some revisions.

- Introduction. The state of the art on the use of supercritical fluid extraction (SFE) as a valid alternative for the green extraction of active compounds from vegetable matter can be enlarged; for this purpose, see this recent review: Baldino et al., Supercritical fluid technologies applied to the extraction of compounds of industrial interest from Cannabis sativa L. and to their pharmaceutical formulations: A review, Journal of Supercritical Fluids, 2020, 165, 104960.

- M&M. CO2 density should be added to Table 1. A SFE plant layout can be added to help the reader in understanding the process.

- Results. The effect of the operative conditions and CO2 density on the extraction performance has to be stressed; in particular, taking into account the variation of CO2 solvent power by increasing pressure and the effect on the selectivity of the process towards the compounds of interest.

Author Response

  1. The state of the art on the use of supercritical fluid extraction (SFE) as a valid alternative for the green extraction of active compounds from vegetable matter can be enlarged; for this purpose, see this recent review: Baldino et al., Supercritical fluid technologies applied to the extraction of compounds of industrial interest from Cannabis sativa L. and to their pharmaceutical formulations: A review, Journal of Supercritical Fluids, 2020, 165, 104960.

We thank the reviewer for the suggestion. We added more explanation on SFE to enlarge the importance (please see page 2).

  1. M&M. CO2 density should be added to Table 1.

We thank the reviewer for the suggestion. The CO2 densities were added to Table 5 because it is more suited under extraction parameters.

  1. A SFE plant layout can be added to help the reader in understanding the process.

We thank the reviewer for the suggestion. The SFE plant layout was added in Supplementary data under Figure S1. An additional change was made in Section 2.1. (page 3) and now states: For a better understanding of the extraction process, a detailed diagram of the supercritical extraction system is shown in Figure S1.

  1. The effect of the operative conditions and CO2 density on the extraction performance has to be stressed; in particular, taking into account the variation of CO2 solvent power by increasing pressure and the effect on the selectivity of the process towards the compounds of interest.

The interconnection between the density and solubility by increasing the pressure was added in the manuscript (see Figure 2) and discussion was updated regarding this process.

Round 2

Reviewer 2 Report

Correction - Comparing the assumed structural formulae (Fig. 4), we may conclude that only Oxepahyperforin - C36H54O5 will be able to lose CO that easily due to oxabicyclo moiety. The consequent loss of 2 prenyl (C5H9) radicals resulting in m/z 413.2350 together with formation of m/z 383.2230 (C24H31O4) ion may occur only for the Furohyperforin, due to specific position of the substituents, hence leaving spectrum b  with most abundant m/z 411.2528 arising due to loss of two pentyl C5H10 radicals for  8-Hydroxyhyperforin 8,1-hemiacetal.

Figure 6. Square-wave voltammograms (SWV) of supercritical CO2 extracts obtained from four medicinal plants: (a) Chamomile; (b) Yarrow; (c) St. John's Wort; (d) Curry plant in the form of microfilm immobilized on the surface of glassy carbon electrode (GCE) and immersed in Britton Robinson buffer (pH 2.5). The experimental conditions are as depicted in Figure 5.

Author Response

Correction - Comparing the assumed structural formulae (Fig. 4), we may conclude that only Oxepahyperforin - C36H54O5 will be able to lose CO that easily due to oxabicyclo moiety. The consequent loss of 2 prenyl (C5H9) radicals resulting in m/z 413.2350 together with formation of m/z 383.2230 (C24H31O4) ion may occur only for the Furohyperforin, due to specific position of the substituents, hence leaving spectrum b  with most abundant m/z 411.2528 arising due to loss of two pentyl C5H10 radicals for  8-Hydroxyhyperforin 8,1-hemiacetal.

We thank the reviewer for the comment and appreciate it very much. We corrected this part as suggested (we added molecular formula, change Fig. 3 to Fig. 4 and correct value for m/z from 41.2528 to 411.2528) in lines 246 – 252.

Figure 6. Square-wave voltammograms (SWV) of supercritical CO2 extracts obtained from four medicinal plants: (a) Chamomile; (b) Yarrow; (c) St. John's Wort; (d) Curry plant in the form of microfilm immobilized on the surface of glassy carbon electrode (GCE) and immersed in Britton Robinson buffer (pH 2.5). The experimental conditions are as depicted in Figure 5.

We thank the reviewer for the comment as well and corrected properly Caption to Fig. 6 (lines 343 – 346).

Reviewer 3 Report

Authors performed the modifications proposed by the Reviewer and improved the manuscript.

Author Response

Thank you once again for your comments.